

# Ensemble decision of local similarity indices on the biological network for disease related gene prediction

Mustafa Özgür Cingiz

Department of Computer Engineering, Faculty of Engineering and Natural Sciences, Bursa Technical University, Bursa, Turkey

Corresponding author
Mustafa Özgür Cingiz,
mustafa.cingiz@btu.edu.tr

## ABSTRACT

Link prediction (LP) is a task for the identification of potential, missing and spurious links in complex networks. Protein-protein interaction (PPI) networks are important for understanding the underlying biological mechanisms of diseases. Many complex networks have been constructed using LP methods; however, there are a limited number of studies that focus on disease-related gene predictions and evaluate these genes using various evaluation criteria. The main objective of the study is to investigate the effect of a simple ensemble method in disease related gene predictions. Local similarity indices (LSIs) based disease related gene predictions were integrated by a simple ensemble decision method, simple majority voting (SMV), on the PPI network to detect accurate disease related genes. Human PPI network was utilized to discover potential disease related genes using four LSIs for the gene prediction. LSIs discovered potential links between disease related genes, which were obtained from OMIM database for gastric, colorectal, breast, prostate and lung cancers. LSIs based disease related genes were ranked due to their LSI scores in descending order for retrieving the top 10, 50 and 100 disease related genes. SMV integrated four LSIs based predictions to obtain SMV based the top 10, 50 and 100 disease related genes. The performance of LSIs based and SMV based genes were evaluated separately by employing overlap analyses, which were performed with GeneCard disease-gene relation dataset and Gene Ontology (GO) terms. The GO-terms were used for biological assessment for the inferred gene lists by LSIs and SMV on all cancer types. Adamic-Adar (AA), Resource Allocation Index (RAI), and SMV based gene lists are generally achieved good performance results on all cancers in both overlap analyses. SMV also outperformed on breast cancer data. The increment in the selection of the number of the top ranked disease related genes also enhanced the performance results of SMV.

## INTRODUCTION

Cancers are mostly polygenic diseases, which are caused by multiple genes that encode proteins unusually (*Sharma & Vella, 2017*; *Tenesa & Haley, 2013*). Essential cellular

functions and biological processes are regulated by proteins and their interactions. Protein-protein interaction (PPI) networks contain relations between proteins, where proteins are presented as nodes and relations between proteins are presented as undirected edges. The perturbation on PPI networks can change optimal structure of biological functions and processes in the cell, which causes various complex diseases (*Chu & Chen, 2008*).

Prioritization of genes from PPI networks are utilized to discover potential disease related genes (*Li et al., 2014*; *Gentili et al., 2022*; *Azadifar & Ahmadi, 2022*; *Luo & Liang, 2015*). The validation of molecular relations in wet lab experiments is an expensive and time consuming work (*Feng, Zhang & Shi, 2020*). PPI networks, which may consist of invalidated interactions, can have high false positive and false negative interactions (*Lei & Ruan, 2013*). Computational techniques lead to discover novel disease related genes on PPI networks. The usage of link prediction (LP) algorithms can construct different types of networks such as social networks (*Gou & Wu, 2022*; *Wang et al., 2023*), PPI networks (*Long et al., 2022*; *Yuen & Jansson, 2023*), drug-disease interaction networks (*Zhang et al., 2022*; *Sadeghi, Lu & Ngom, 2022*), metabolic networks (*Ekkers et al., 2022*), scientific collaboration networks (*Lande et al., 2020*; *Kim & Diesner, 2019*), and other type of homogeneous or heterogeneous networks (*Lim, Jhanjhi & Abdullah, 2019*; *Lim, Abdullah & Jhanjhi, 2021*).

*Kumar et al. (2020)* classified LP algorithms into four main categories, which were similarity based algorithms, probabilistic and maximum likelihood based (PML) algorithms, dimensionality reduction based (DRB) algorithms, supervised based algorithms. Early studies of LP algorithms asserted that if two genes in PPI networks share similar topological features, they are likely to interact with each other (*Yang et al., 2015*). Similarity based algorithms exploit node, path or hybrid of node and path similarity metrics to predict new, missing or spurious links (*Mutlu et al., 2020*). Node-based similarity algorithms are called local similarity indices or common neighbors-based algorithms (*Kumar et al., 2020*; *Wu et al., 2022*) hypothesize that more common neighbors between nodes lead to a greater tendency for association between these nodes. Local similarity indices can also take into account the degree of nodes. Common neighbors (CN), the Jaccard Index (JC), Adamic-Adar (AA), Preferential Attachment (PAC), Resource Allocation Index (RAI), the Sorensen Index, the Salton Index, CAR-based indexes, the Hub Promoted Index, and the Leicht–Holme–Newman Local Index are some commonly used local similarity indices (*Mutlu et al., 2020*; *Lü & Zhou, 2011*). Path based similarity techniques, which are also called global similarity indices, intend to solve the problem of local based approaches, which may not capture whole topological features of complex networks. Global similarity indices take into account paths between nodes on entire network, which means the detection of more high level topological relations between nodes. The Katz Index, Local Path Index, Global Leicht–Holme–Newman, Local Random Walk, Random Walk with Restart, Average Commute Time, and SimRank are well known global similarity indices (*Yang et al., 2015*; *Lü & Zhou, 2011*; *Martínez, Berzal & Cubero, 2016*). However, global similarity indices have target to overcome the problems of local similarity indices, global similarity indices are not too feasible to apply on large networks

due to their complexity (*Lü & Zhou, 2011*). Quasi-local similarity indices intend to utilize the advantages of local and global similarity indices *via* effective trade-off between them (*Kumar et al., 2020*). Local path index, path of length, similarity based on local random walk and superposed random walk are popular quasi-local similarity indices (*Kumar et al., 2020*; *Lü & Zhou, 2011*). PML algorithms build parameter based model, which were designed to optimize objective function. Parameters are derived from network data and the probability of new links between nodes are calculated due to parameters of distribution (*Wu et al., 2022*). DRB algorithms map graph and its features into lower dimensional space. Extensive graphs such as PPI networks are required to overcome the curse of high dimensionality, which is intended to solve by DRB algorithms. DRB algorithms can be classified into two main categories: embedding based and matrix factorization based DRB algorithms (*Kumar et al., 2020*). Embedding based DRB algorithms encode network structure into embedding space (*Perozzi, Al-Rfou & Skiena, 2014*; *Grover & Leskovec, 2016*). Matrix factorization based DRB algorithms reduce dimension of complex networks and generate latent space by using techniques such single value decomposition, non-negative matrix decomposition (*Yang et al., 2022*). Supervised learning algorithms (*Kumari et al., 2022b*; *Malhotra & Goyal, 2021*) identify features of node pairs and assign a positive label if the nodes are related. Support vector machines, k-nearest neighbor, naive Bayes, artificial neural network based classifiers are popular methods in supervised learning based link prediction algorithms (*Wu et al., 2022*; *Kumari et al., 2022b*; *Malhotra & Goyal, 2021*).

Genome-wide associations studies (GWAS) focus on thousands of genetic variants across many genomes to discover disease related locus, which may present potential disease related genes (*Uffelmann et al., 2021*). Online Mendelian Inheritance in Man (OMIM) (*Hamosh et al., 2000*) and DisGeNet (*Piñero et al., 2020*) are GWAS based databases, which contain relations between genes and diseases. However, GWAS are important for detecting disease related genes, they are time consuming and expensive. LP algorithms are fast and effective to derive candidate genes for diseases (*Lan et al., 2015*; *Madeddu, Stilo & Velardi, 2020*).

The main objective of the study is to infer candidate disease related genes by using local similarity indices by rank based approach separately and also applying ensemble technique on the same similarity indices to derive more robust disease related genes. The study intends to measure how much a simple ensemble decision of LSIs affects the performance of the inferred genes. I applied four LSIs for LP to prioritize and discover disease related genes from a PPI network, the Human Protein Reference Database (HPRD). Gastric, colorectal, breast, prostate and lung cancers related genes were retrieved from the basic OMIM database. Four different local similarity indices scores between disease related genes and other genes were derived from the PPI network. The top 10, 50 and 100 disease related genes were determined due to ranking of local similarity indices scores in descending order. The inferred disease related genes of five cancers were validated using the GeneCard dataset, whose number of disease related genes are more than the basic OMIM database. GeneCard was utilized in overlap analysis for the validation. The performance of each local similarity indices was evaluated in overlap analysis. The

ensemble techniques can enhance the performance of disease related gene predictions with integration of different LP based disease related gene predictions. Simple majority voting (SMV) was employed as an ensemble technique on LSIs to derive more accurate and reliable disease related genes for five cancer types. SMV identified the top 50, 100 and 200 disease related genes by considering the genes that appeared at least twice in the separate lists of the top 50, 100 and 200 disease related genes generated by the four LSI techniques. The biological assessment was also employed by using Gene Ontology (GO) terms to evaluate LSIs and SMV based inferred gene lists. The another objective of the study is to assess the validity of ensemble-based predictions through GO-term analysis. GO-terms analysis can help researchers to perceive disease related biological process and functions to understand underlying mechanisms of diseases. The disease related GO-terms, which were derived by using OMIM disease related genes in gene set enrichment analysis (GSEA), were utilized as the validation data in the biological assessment.

The contributions of the study can be summarized as:

- The integration of disease related gene predictions based on LSIs is not well studied area in LP. The effect of the integration of the predictions of LSIs based on a simple ensemble decision technique for the disease related gene predictions was investigated.
- The success of the inferred gene lists of five cancers were evaluated to understand how they were related to cancers based on GO-terms.
- The performance of LSI based gene lists and SMV based gene lists on five cancers were investigated separately. Their results were also compared to the results of studies in the literature.

The rest of the article is organized as the following: "Related Works" summarized the similar studies about LP in the literature. "Materials and Methods" presented materials and methods part, which includes datasets in the study, LSIs and the proposed method. "Results" depicted the performance results of the LSIs based gene lists and SMV based gene lists in overlap analysis. "Discussion" summarized the targets and the findings of the study. "Discussion" also compared the results of the study to similar studies in the literature and also presented the limitations of the study.

## RELATED WORKS

LSIs were exploited in the predictions of molecular relations on many biological networks. These studies generally tried to expand candidate disease related genes or build biological networks with new or missing link predictions. LSIs are also easy to implement in LPs; thus, they can be utilized for ensemble decision to determine more robust link predictions.

*Ghiassian, Menche & Barabási (2015)* ranked genes based on their relationship to disease-specific genes, which were part of disease-related modules, by taking advantage of the interactome's interconnection properties. They utilized OMIM and PheGenI databases for the prediction validation. Diffusion models generally intended to expand candidate disease related genes on PPI network *via* random walk and random walk variant models on graph (*Mutlu et al., 2020*; *Wu et al., 2022*; *Lü & Zhou, 2011*). They also utilized GWAS

based databases such as OMIM and DisGeNET databases in prediction analysis (*Piñero et al., 2020*; *Madeddu, Stilo & Velardi, 2020*). The most of studies in the literature evaluated the link prediction algorithms performance *via* precision, recall, F-measure values, AUC scores on various complex networks such as the network of Jazz musicians, food web, the carbon exchange network, email network, Facebook, PPI networks, the network of USA airline (*Liu et al., 2017*; *Kumari et al., 2022a*). When comparing the performance results of the gene lists based on LSIs and their ensemble inference in this study with those from other studies in the literature, the results were similar. *Liu et al. (2017)* applied LP algorithms to predict the missing links in 15 different complex networks. In the study (*Liu et al., 2017*), four local indices and four global indices were applied for the missing link prediction. The study divided the original datasets into 90% train data and 10% test data for the cross validation to calculate precision and other performance metrics. *Kumari et al. (2022a)* used four global similarity indices to determine relations between nodes to build communities. The relation predictions of five different communities were combined to obtain final community for four real networks. *Zhao et al. (2011)* ranked candidate disease genes using gene expression data and the PPI network. True positive rates (TPR), false positive rates (FPR) and receiver operating characteristic (ROC) metrics were utilized for the performance evaluation in their study. Another comprehensive study (*Martínez, Berzal & Cubero, 2016*) applied local, global and quasi-local similarity indices to make comparison of the performances of different metrics on seven complex networks, which also contain protein-protein interaction network of budding yeast (YST). This extensive survey emphasized the strength of local similarity indices in complex network. The recent studies (*Kumar et al., 2020*; *Kumari et al., 2022a*) worked on complex networks, which did not cover PPI networks, also analyzed the link prediction performance results using precision, recall and F-measure values. *Lee & Tukhvatov (2018)* exploited LSIs on social media network, VKontakte, to compare the local similarity metrics *via* precision, recall and F-measure performance evaluation metrics. *Kumar et al. (2020)* worked on the commonly used complex networks, which some of them were also used in *Liu et al.*'s *(2017)* work, and he found significantly lower precision and recall values.

*Zhang, Tong & Wu (2020)* developed a novel linear model for integrating various types of LSIs and employed two typical model-averaging approaches, which were typical model estimators for the model selection, Akaike information criteria, and Bayesian information criteria. *Zhang, Tong & Wu (2020)* applied their proposed model with LSIs in six well known complex networks. Their selection method slightly outperformed in complex networks. *Chiu & Zhan (2018)* constructed an input feature vector by utilizing LSIs. Chiu applied a stochastic learning weak estimator (SLWE) tried to estimate link existence probabilities between nodes in dynamic networks, where nodes and edges were inferred by LSI based feature vector. If a node was inferred by any LSI, then it was labeled as 1 by SLWE for link prediction classification in the study. This study investigated the link prediction in dynamic networks and employed the proposed model in three networks and to evaluate the performance of the proposed model the studies used precision, recall, and AUC scores. *Wu et al. (2019)* employed the Ordered Weighted Averaging (OWA) operator as a serial ensemble strategy to enhance link prediction performance with using nine LSIs.

Wu ranked nine LSIs based on precision scores of their predictions and some LSIs based predictions were eliminated by OWA operator. *Qiu et al. (2020)* utilized LSIs to extract edge features from two bitcoin datasets to develop tree based classifiers to determine accurate relations in directed graph.

## MATERIALS AND METHODS

The datasets of the study, LSIs, and the proposed methodology were introduced in materials and methods section.

### Datasets

The Human Protein Reference Database (HPRD, Release 9), Online Mendelian Inheritance in Man (OMIM), the GeneCard dataset, and GO-terms were exploited in this study. The HPRD database (*Keshava Prasad et al., 2009*) consists of 8,603 unique proteins which have 44,376 protein-protein interactions. HPRD presents post-translational modifications interaction networks and disease association for each protein in the human proteome (*Keshava Prasad et al., 2009*). After the advancements in high throughput DNA sequencing technology, single nucleotide polymorphisms (SNPs) across humans can be revealed by GWAS. GWAS for human use the entire genome of hundreds of thousands of humans to detect disease causing mutations (*Hamazaki et al., 2017*). The OMIM database (*Hamosh et al., 2000*) was created using GWAS in which many human disease-gene relations are presented. EnrichR (*Kuleshov et al., 2016*) is an online gene set enrichment analysis tool, which also contains a collection of biological databases such as the OMIM database. OMIM database was directly generated from the NCBI's OMIM Morbid Map in EnrichR (*Chen et al., 2013*).

To validate the inferred gene list, this study utilized the GeneCard (*Safran et al., 2010*) dataset of gene-disease relations, which was constructed using different bioinformatics databases from the literature. GeneCard retrieved disease-associated genes from NCBI and UniProt, as well as disease related genes was also obtained using gene expression data and GWAS data. GeneCard disease-gene list contains 103, 135, 228, 91, 131 disease-related genes for gastric, colorectal, breast, prostate and lung cancers, respectively. I also evaluated the performance of the LSIs and SMV based gene lists using the GO-terms, which were retrieved by g.Profiler python package (version 1.0.0) (*Kolberg et al., 2023*).

### Local similarity indices

A graph can be represented as G = (V, E), where V is a set of nodes and E is a set of edges indicating relations between nodes. Proteins are nodes and relations between proteins are edges in PPI networks. LSI-based methods take into account the topological structure of graph, which may be such as number of common neighbors, node degree, the shortest distance of nodes in PPI network, to predict new or missing link between nodes. The study presented the set of adjacent nodes of $V_x$ as $\Gamma(V_x)$, which indicates the number of neighbors of node x. The degree of node, $V_x$, was given as $k_x$, and the similarity score between node x and node y was also presented as S(x, y) in the notation of the article. Preferential Attachment (PAC), the Jaccard Index (JC), Adamic-

Adar (AA), and the Resource Allocation Index (RAI) were utilized as LSIs to predict potential relations between nodes.

### Preferential Attachment Index

Biological networks generally present scale free topology, where many of nodes have few neighbors and few of nodes have numerous neighbors. The degree of nodes in scale free topology networks follow a power low distribution. The Preferential Attachment Index (PAC) (*Barabási & Albert, 1999*; *Xie, Zhou & Wang, 2008*) assumes that if a node has high degree than it has tendency to interact to other nodes. PAC leads to have less number of probable relations for nodes, which have lower degree. PAC is a favored metric due to its simplicity and computational time.

$$S(x, y)^{PAC} = k_x k_y \tag{1}$$

PAC based similarity score calculation is given in Eq. (1). PAC based similarity score is obtained *via* multiplication of degree of node x, $k_x$, and node y, $k_y$.

### Adamic-Adar Index

The high number of common neighbors of two nodes with low degree values increases the value of the Adamic-Adar based similarity score, which was first employed by *Adamic & Adar (2003)* for comparison of web pages. The Adamic-Adar Index (AA) score is logarithmically penalized due to the degree of common neighbor nodes. The calculation of AA based similarity score between node x and node y is presented in Eq. (2).

$$S(x, y)^{AA} = \sum_{z \in \Gamma(x) \cap \Gamma(y)} \frac{1}{\log k_z} \tag{2}$$

The AA based similarity score utilizes common neighbor nodes of two nodes, x and y, which are presented as z is Eq. (2). The high degree of common neighbor nodes, $k_z$, decreases the AA similarity based score.

### Jaccard Index

The Jaccard Index (JC) is a normalized form of common neighbor metric (*Jaccard, 1901*). JC assumes that if the ratio of the number of common neighbors of node x and node y to all neighbors of the two nodes is close to one, JC based similarity score become higher. Equation (3) presents the calculation of JC based similarity score between node x and node y.

$$S(x, y)^{JC} = \frac{|\Gamma(x) \cap \Gamma(y)|}{|\Gamma(x) \cup \Gamma(y)|} \tag{3}$$

### Resource Allocation Index

The Resource Allocation Index (RAI) similarity score is similar to AA similarity score, where they differ from each other *via* penalization. RAI similarity score between node x and node y is also calculated using common neighbor nodes with their degree values. RAI does not apply the logarithmic penalization, which leads to punish the high-degree

common neighbors more heavily than AA (*Ou et al., 2007*; *Zhou, Lü & Zhang, 2009*). RAI similarity score calculation is presented in Eq. (4).

$$S(x, y)^{RAI} = \sum_{z \in \Gamma(x) \cap \Gamma(y)} \frac{1}{k_z} \qquad (4)$$

However, AA and RAI performed similar performance results on smaller size networks, RAI outperformed on networks, where it contains high degree of nodes (*Kumar et al., 2020*).

## Simple Majority Voting

Simple Majority Voting (SMV) is a well-known ensemble learner, which intends to integrate predictions of different methods (*May, 1952*; *Hayat et al., 2022*). SMV assigns the equal weights, which is equal to one, for predictions of methods separately. If half or more than half of the number of the methods infer any prediction, SMV determines the prediction as a valid prediction. Predictions made by less than half the number of methods are eliminated by SMV. The weights of SMV based prediction is two, which is the half of the number LSIs in the study. If the total weight of a prediction is two or more than two, the prediction is added into SMV based prediction list. Predictions made by less than half the number of methods are eliminated by SMV.

## The proposed system

The discovery of disease specific genes is essential to develop disease related drugs for disease treatment. The study intends to measure how well LSIs and ensemble decision based results of LSIs predict disease related genes. First HPRD, a PPI database, was exploited to build a PPI network as presented in Fig. 1.

PAC, AA, JC and RAI similarity scores between all genes, which are presented as nodes in HPRD network, were calculated. HPRD network contains all types of genes some of them are related to disease and others are not. Each gene pair has four similarity scores, which indicate their potential relations. If a gene pair has a high similarity score, there is a likely relation between these two genes. After calculating the similarity scores, the gene pairs were ranked separately in descending order according to their four local similarity scores.

The study hypothesized that genes with a high local similarity score with any disease related genes associated with the disease. The basic OMIM database was exploited to obtain disease related genes which are presented in Table 1.

The basic OMIM database contains 11, 40, 28, 30 and 21 gastric, colorectal, breast, prostate and lung cancers related genes respectively. The probable disease related genes were determined based on their scores across four LSIs, along with all disease-related genes, as presented in Table 1. To clarify, if a gene has the top 10, 50 or 100 highest score between any disease related genes then it is selected for the disease specific gene list. This selection procedure led to infer 12 disease-related gene lists for five diseases separately due to the multiplication of the number of the top selection criteria, which are 10, 50, and 100, and the number of LSIs which are AA, PAC, JC, and RAI. A total of 60 different gene lists were created due to multiplication of the number of disease, the number of local similarity scores and the number of top selection choices.

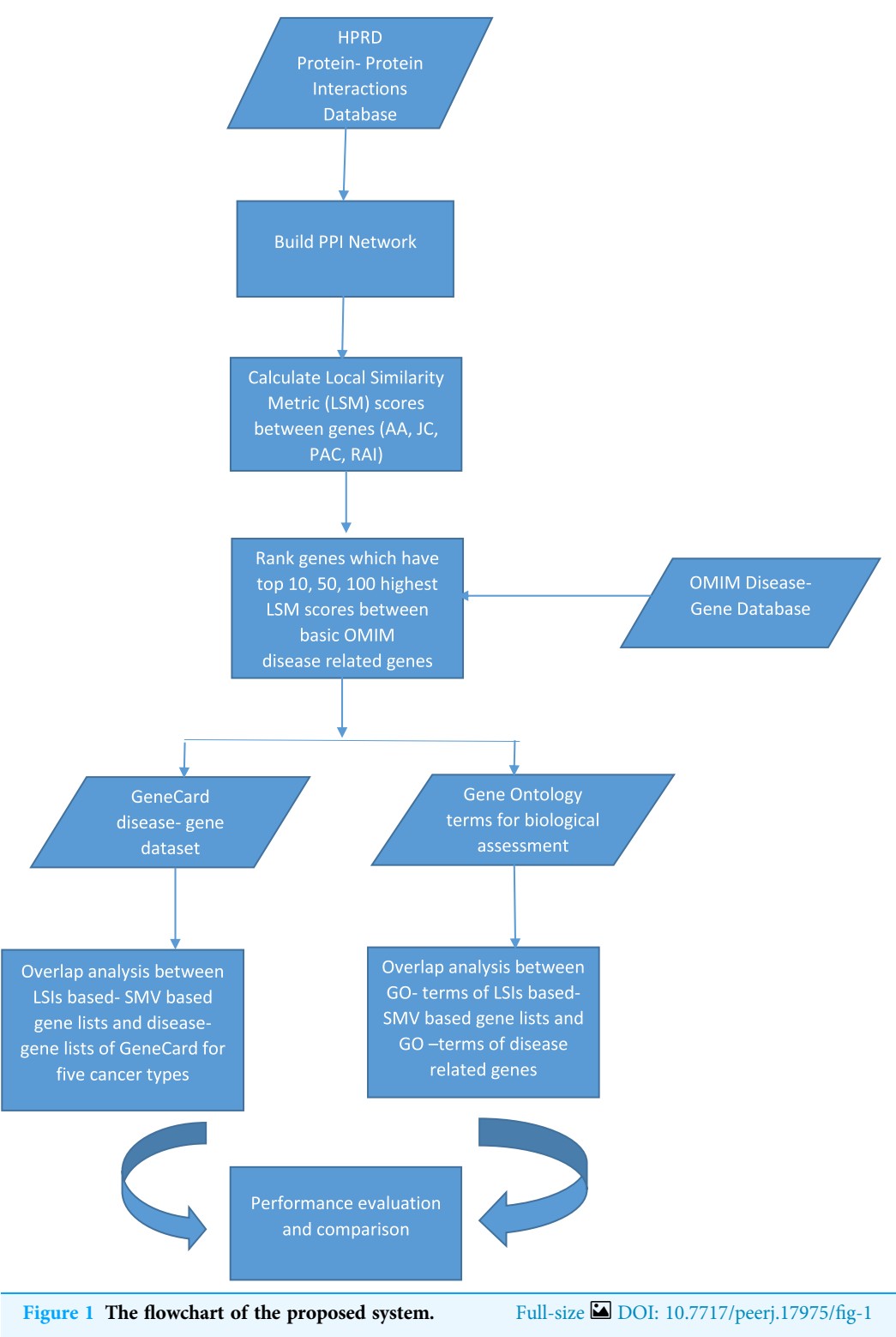

**Figure 1 The flowchart of the proposed system.**

In addition, I applied a simple ensemble technique, SMV, utilizing the scores of LSIs on gene lists of five diseases, which are determined according to the top 10, 50, or 100 highest local similarity scores separately. SMV determines SMV based disease related genes for

**Table 1 Diseases and disease related genes.**

| Disease name | Disease related genes |
|---|---|
| Gastric cancer | IL1RN, IL1B, IRF1, KLF6, APC, PIK3CA, CASP10, CDH1, ERBB2, MUTYH, FGFR2 |
| Colorectal cancer | CRCS6, CRCS7, DCC, MLH1, CRCS5, CRCS2, PLA2G2A, AXIN2, BRAF, MSH2, SMAD7, MLH3, MCC, TGFBR2, CRCS8, PDGFRL, CRCS9, TLR2, HMPS1, APC, PIK3CA, DLC1, BUB1B, MSH6, BAX, TLR4, FGFR3, TP53, CTNNB1, CRCS11, CRCS10, FLCN, NRAS, CCND1, EP300, CHEK2, AKT1, BUB1, PMS2, PMS1 |
| Breast cancer | NQO1, XRCC3, BRCD2, PALB2, ESR1, RAD51A, PPM1D, PIK3CA, ATM, SLC22A1L, TP53, KRAS2, TSG101, PHB, HMMR, RB1CC1, BRCA2, BRCA3, BRIP1, CASP8, RAD54L, CDH1, CHEK2, BRCD1, AKT1, BRCATA, BCPR, BARD1 |
| Prostate cancer | MSR1, ZFHX3, ELAC2, HPCQTL19, AR, EHBP1, KLF6, HPC3, HPC5, HPC4, HPC7, HPC6, HPC9, MAD1L1, HIP1, HPC11, HPC10, RNASEL, PCAP, CD82, HPC15, HPC14, HPCX2, PTEN, BRCA2, HPCX1, MXI1,CHEK2, EPHB2, MSMB |
| Lung cancer | TSG11, CHRNA3, CHRNA5, DDX26, MPO, BRAF, DLEC1, LNCR1, EGFR, RASSF1, IRF1, LNCR4, CASP8, LNCR3, CYP2A6, PIK3CA, PPP2R1B, SLC22A1L, ERCC6, MAP3K8, KRAS2 |

each disease using genes that are inferred by at least two LSI based gene lists. The threshold value is determined as two due to the half of number of local similarity metrics, which is equal to four. For instance, if gene X is in at least two of four LSIs based gene lists related gastric cancer, which were created by the top 10 local similarity scores, SMV adds gene X for SMV based gene list related to gastric cancer that is formed due to the top 10 local similarity scores. There were 15 SMV based gene lists were formed due to the multiplication of the number of diseases, five, and the number of top selection choices, three.

Figure 2 depicted the pseudocode of SMV based link prediction. OMIM disease gene list, other genes out of from the OMIM disease gene list and the number of LSI techniques were presented R, O, and I respectively in Fig. 2. Algorithm begins with the inference of PPI network from HPRD database. Four LSI techniques determine the similarity scores between genes of R and O lists, which are presented as $LP_{indices,i}$, to measure the association between potential disease related genes and disease related genes on HPRD network. All four LSI techniques based predictions were ranked in descending order using RA rank function in Fig. 2. LSI techniques based ordered lists were aggregated into $LP_{final}$ separately. The number of ranked genes, rv, was also presented 10, 50 and 100 in Fig. 2. The last loop investigated how many times potential disease related genes were involved in $LP_{final}$, where if the number of any gene was equal to or greater than two, it was added into SMV based prediction list.

The study inferred 60 separate gene lists and 15 SMV based gene lists using LSIs for five diseases. Overlap analysis was applied to evaluate the performance of the inferred gene lists. Overlap analysis with the GeneCard dataset and biological assessment were exploited to evaluate disease related gene lists.

## RESULTS

First, the overlap analysis with the GeneCard dataset and biological assessment with GO-terms were exploited in the performance evaluation part. The performance of the LSIs-based and SMV-based gene lists in the overlap analysis and biological assessment parts was evaluated using the performance metrics. This chapter started by outlining the

**Input**:

HPDR: Human Protein Interaction Database

$O$= OMIM disease gene list, $o$= |O| the number of genes in O

$R$= Other genes, not belong to the O list, $r$= |R, the number of candidate genes

$I$, the number of LSI techniques, which is four in the study

$rv$, rank value $rv \in \{10, 50, 100\}$

$LP_{indices,i}$   LSI scores for each LSI techniques, $i \in$ {PAC, AA, JC, RAI}

LSI (net, i,k,p) Calculation of LSI score between gene $k$ and gene $p$ by LSI technique $i$ on net

$RA$, is rank function in descending order

$RLP_{indices,i}$   Ranked list of LSI, $i \in$ {PAC, AA, JC, RAI}

$RLP_{final}$   Aggregation of all ranked list of LSIs, $i \in$ {PAC, AA, JC, RAI}

$SMV$, SMV based predictions

**Output**:

SMV, SMV based interactions list

```
begin:
    net = create PPI network from HPRD
    RLP_final = {}
    for i=1 to len (I) do
        for j=1 to len (R) do
            for k=1 to len (O) do
                LP_indices,i = LSI (net, i, j, k)
            end
        end
    end
    for i=1 to len (I) do
        RLP_indices,i = RA(LP_indices,i)
        RLP_final += RLP_indices,i
    end
    for i=1 to rv do
        if gene_x in RLP_final ≥ 2
            add gene_x into SMV
        end
    end
end
```

**Figure 2  Pseudocode of SMV based link prediction.**

performance metrics. The study then reported the performance findings from the GeneCard overlap analysis and the biological evaluations of gene lists based on SMV and LSIs, respectively.

## Performance metrics

Overlap analysis evaluated the performance of the inferred gene lists *via* the GeneCard dataset, which was exploited as the validation dataset. If a gene in LSI based inferred gene
list for the specific disease is also in the GeneCard dataset, which contains the specific disease-gene relations, then it was determined as a true positive (TP) in overlap analysis. Otherwise, it was labeled as false positive (FP). If a gene in the GeneCard dataset for the specific disease is not inferred by a LSI based gene list for the same disease, then it was labeled as false negative (FN) in overlap analysis. Precision, recall and F-measure metrics are the well-known performance evaluation metrics, which were exploited in the overlap analysis of the study. The calculation of three evaluation metrics was presented in Eqs. (5)–(7). The genes in the GeneCard gene list is limited, which may lead to retrieve low TP values. This situation led to obtain the low performance results in overlap analysis.

$$Precision = \frac{TP}{TP + FP} \tag{5}$$

$$Recall = \frac{TP}{TP + FN} \tag{6}$$

$$F - measure = 2 * \frac{Precision * Recall}{Precision + Recall} \tag{7}$$

The biological assessment also applied overlap analysis by using GO-terms. First, disease related GO-terms, which were retrieved by using OMIM disease gene list, were exploited as validation data of the biological assessment. If a GO-term was inferred by LSI based or SMV based gene list in GSEA and it is also in GO-terms in validation set, it was determined as TP in the biological assessment based overlap analysis. Otherwise, it was labeled as FP. If a GO-term in validation set is not inferred by LSI-based or SMV based gene lists in GSEA, then it was determined as FN in the biological assessment.

ROC curve is a visual representation technique to assess the classification performance of models across various threshold values. ROC curve also provides a successful performance evaluation in classification processes that are difficult to evaluate due to class distribution and imbalanced data problem. ROC curves can be used to compare how well various approaches perform in terms of sensitivity and specificity when evaluating the results of the overlap analysis. Sensitivity, recall or true positive rate (TPR) is the ratio of TP to all actual positive samples. Specificity is the ratio of true negative (TN) to all actual negative samples. ROC curve utilizes TPR and false positive rate (FPR), which is equal to 1-specificity, to plot ROC curves. The area under the curve (AUC) score indicates how well classification was performed. If AUC score is close to one, then the performance of model is successful. The high AUC score indicates high TPR and low FPR values across different threshold values, which determine a high proportion of actual positives and a low proportion of actual negative samples as positives samples.

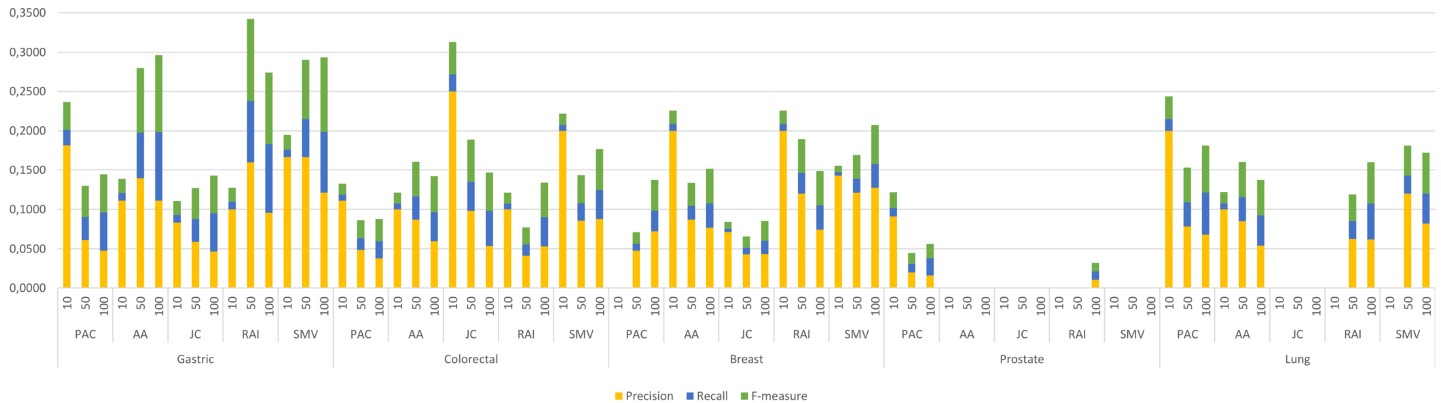

**Figure 3 Individual performance evaluation of LSI and SMV based genes.**

## The performance evaluation of the LSI-based and SMV based gene lists using the GeneCard dataset in overlap analysis

The four LSI values for all possible gene pairs on HPRD PPI network were derived as a first step in the study. The study focused on the relations of disease related genes, which are in basic OMIM gene-disease gene database. If a gene has high LSI score with any disease related gene from basic OMIM database by ranking, then it is regarded as potential disease related gene. In this section, the study evaluated the performance of LSI based inferred genes, which have relations with basic OMIM gene-disease list. Figure 3 presented the individual performance results of LSI based and SMV based inferred genes in overlap analysis.

The stacked bar chart in Fig. 3 presented the precision, recall and F-measure scores of the top 10, 50 and 100 inferred genes by LSIs in overlap analysis for five cancer types. The worst performance results in overlap analysis were generally obtained from JC similarity based genes in four of five cancer types. JC does not apply any penalization like RAI and AA, which may increase the impact of hub genes and lead to poorer outcomes. The performance outcomes of LSIs-based genes in prostate cancer exhibited the lowest values among all cancer types. However, the performance results on gastric cancer were generally higher than the performance results of other cancers. The highest performance results were achieved by RAI, JC, SMV based inferred genes on gastric, colorectal, and breast cancer respectively. PAC based inferred genes outperformed on prostate and lung cancers. With the exception of prostate cancer, it was observed that the F-measure values increased as the number of genes included in the overlap analysis increased. The highest precision values were also retrieved when choosing the top 10 inferred genes. The precision value significantly increased if any of the top 10 LSI-based predicted genes were also found in the GeneCard dataset. Based on the performance results derived from SMV, an analysis of F-measure values revealed that, with the exception of prostate cancer, SMV was in the top two most successful algorithms across all four cancer types. The all numeric values of overlap analysis with GeneCard were given in Table S1.

The study also depicted the effect of the ensemble approach by comparing the best and worst performance results obtained from LSI-based methods with the performance results
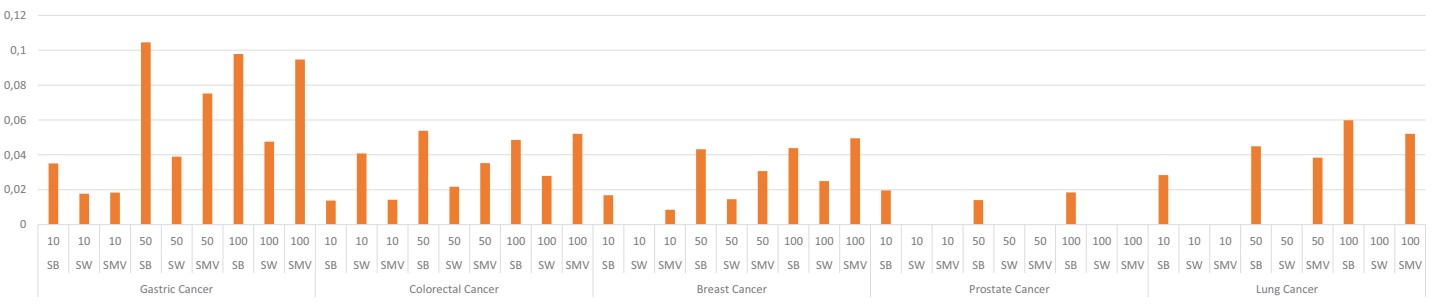

**Figure 4 F-measure values of single best (SB), single worst (SW) and SMV results.**

from SMV in Fig. 4. SMV utilized four LSIs-based genes, thus the performance results of SMV were expected to range between SB and SW in overlap analysis. The highest F-measure values according to the genes selected by the ranking of the top 10 and 50 genes were determined with SB based genes. However, the highest F-measure values were obtained from SMV based genes on breast cancer when taking into account the top 100 ranked genes. Furthermore, the highest F-measure values of SB based genes were too close to results of SMV based gene lists on breast cancer when the number of the top ranked genes was determined as 100. These findings were important for emphasizing the efficiency of the ensemble approach. The increment in the number of ranked genes also enabled SMV to predict more reliable and accurate disease related genes.

The performance of the overlap analysis between disease-related genes and LSI-and SMV based genes for each cancer type was also evaluated with ROC curves in Fig. 5. The study employed only three threshold values, which were top 10, 50, and 100 ranked genes, to measure the performance of each LSI-based gene list for five cancers. The optimal ROC curves could not be plotted efficiently due to the limited threshold values and low performance results in overlap analysis by using the GeneCard dataset. Figure 5 depicted that all AUC scores under ROC curves of LSI-based and SMV based genes in overlap analysis was close to 0.5, which indicated that the success of the genes obtained with the LSI-based and SMV based gene lists in the overlap analysis was close to the success of the random prediction.

The highest AUC scores were retrieved by the inferred gene list on gastric cancer. SMV did not significantly enhance the performance of overlap analysis for all five cancer types. The results of the overlap analysis with the GeneCard dataset emphasized that high FP, FN, and TN values led to have low AUC scores under ROC curves.

## The performance evaluation of the LSI-based and SMV based gene lists using GO terms

Genes associated with the same disease have tendency to be part of similar biological processes, biological functions and sub-cellular localizations (*Cingiz, Biricik & Diri, 2021*; *Cao et al., 2024*; *Zhang et al., 2024*). The GO project comprises three structured ontologies that categorize gene products based on their biological processes, molecular functions and cellular components. The GO represents product attributes through specific terms called GO-terms. GO-terms group genes involve in the same biological processes, activities or
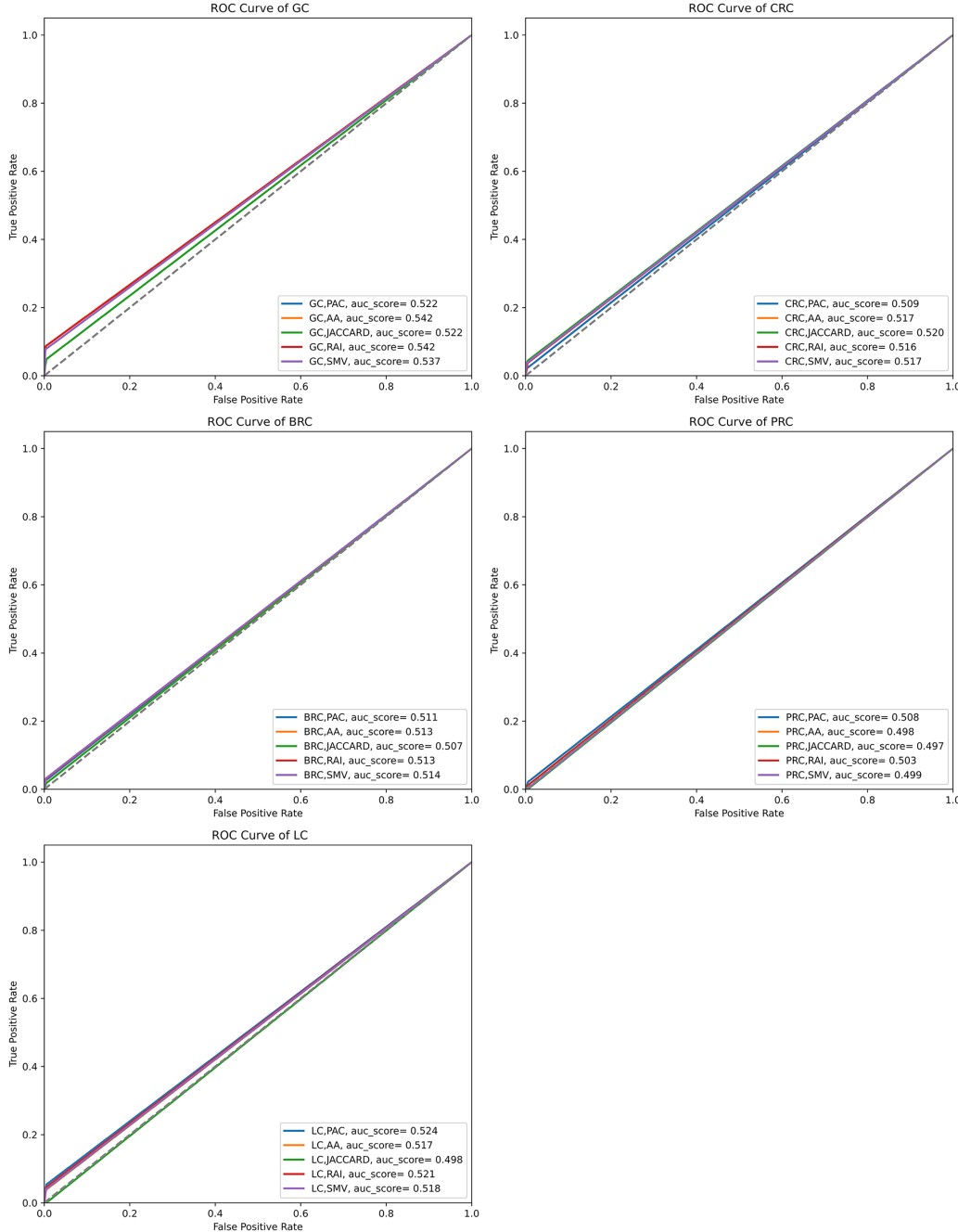

**Figure 5  ROC curve analysis of LSIs and SMV based inferred genes in overlap analysis.**

cellular structures. The biological process and molecular function ontology terms provide insight into the activities or events that a gene product is involved in. Cellular component terms specify the location within a cell where a gene product functions. The performance of LSIs-based and SMV based gene lists were assessed by utilizing GO-terms. Figure 6 presented how GO-terms were used for the performance evaluation of inferred gene lists.

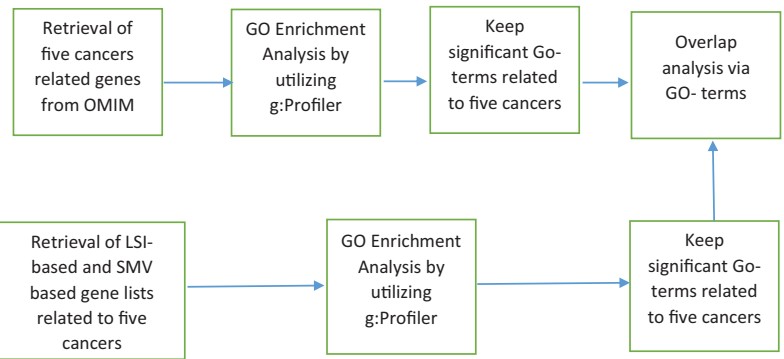

**Figure 6** LSI-based and SMV based gene lists using GO terms.

I utilized the OMIM disease gene list, which was also employed to infer the LSIs-based gene list for five cancers on the HPRD dataset, to identify significant cancer related GO-terms. The g: Profiler python package (*Kolberg et al., 2023*) (version 1.0.0) was applied to identify significant GO-terms from the five cancer gene lists through GSEA. In g: Profiler, *p*-values were adjusted using the Bonferroni correction method, with an adjusted *p*-value threshold of <0.05, which was also used as threshold value in the study. After the identification of significant disease related GO-terms, LSI-based and SMV based gene lists were exploited to determine the GO-terms of inferred gene lists by LSIs and SMV. GSEA was also employed to derive the GO-terms of all LSI-based and SMV-based gene lists. The GO terms from LSI-based and SMV-based gene lists were evaluated by comparing them with disease-related GO terms derived from the OMIM disease gene list in overlap analysis. If a GO-term of LSI or SMV base gene list was also in GO-terms of disease-related GO-terms, it was labeled as TP. Otherwise, it was labeled as FP. If a disease related GO-term was not overlapped in GO-terms of LSI or SMV base gene list, then it was labeled as FN. The study evaluated the performance of GO-terms of LSIs-based and SMV based gene lists using precision, recall, F-measure, and ROC curves. Figure 7 depicted the performance of GO-terms of LSIs-based and SMV based gene lists.

The performance results of LSIs and SMV based gene lists by using GO-terms, which can be accessible in Table S2, presented higher than the performance results of LSI and SMV based gene lists by using the GeneCard dataset. RAI outperformed on gastric, colorectal, and lung cancer data when the top number of was selected as 50 or 100. RAI achieved more than 0.5 F-measure values, which were 0.57, 0.62, and 0.54 for gastric, colorectal, and lung cancer respectively. The performance results of AA were similar to those of RAI across four cancer types, except in the case of prostate cancer, where AA achieved the highest F-measure score, 0.47, when the top-ranked value was set to 100. SMV presented the highest F-measure score, 0.52, on breast cancer if the top 100 ranked genes were selected. The performance of LSIs and SMV based gene lists were also higher on gastric and colorectal cancer data. The SMV based gene lists demonstrated performance close to the highest values across all cancer types. The increment of the top ranked genes in overlap analysis of GO-terms also enhanced the performance results. The lowest F-

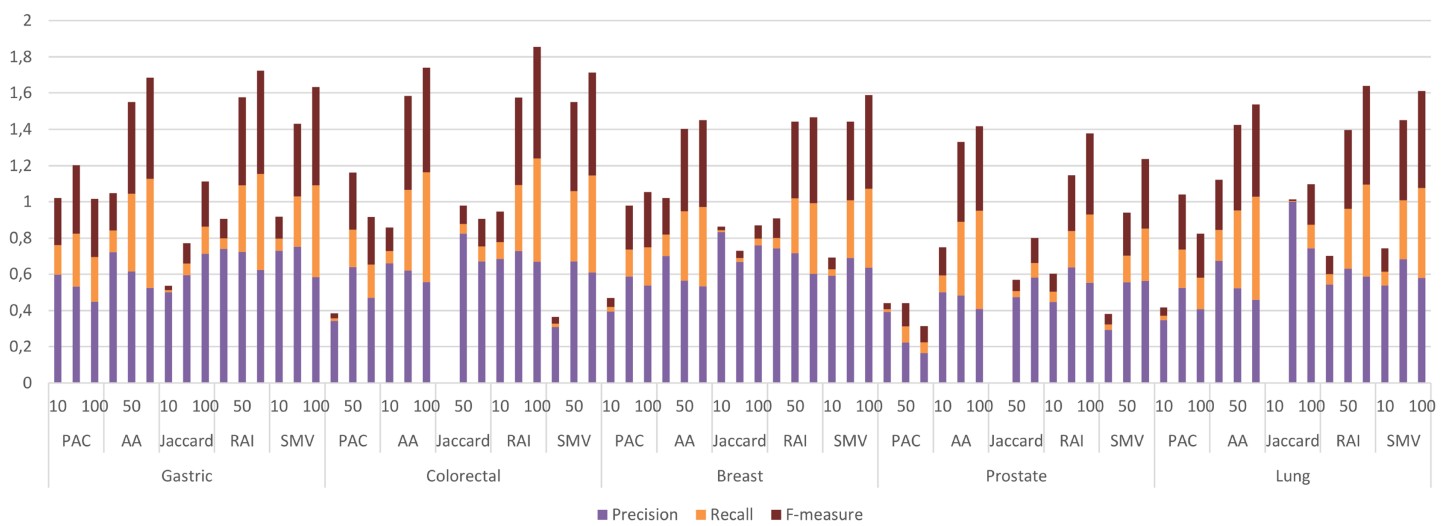

**Figure 7 Individual performance evaluation of LSI and SMV based genes using GO-terms.**

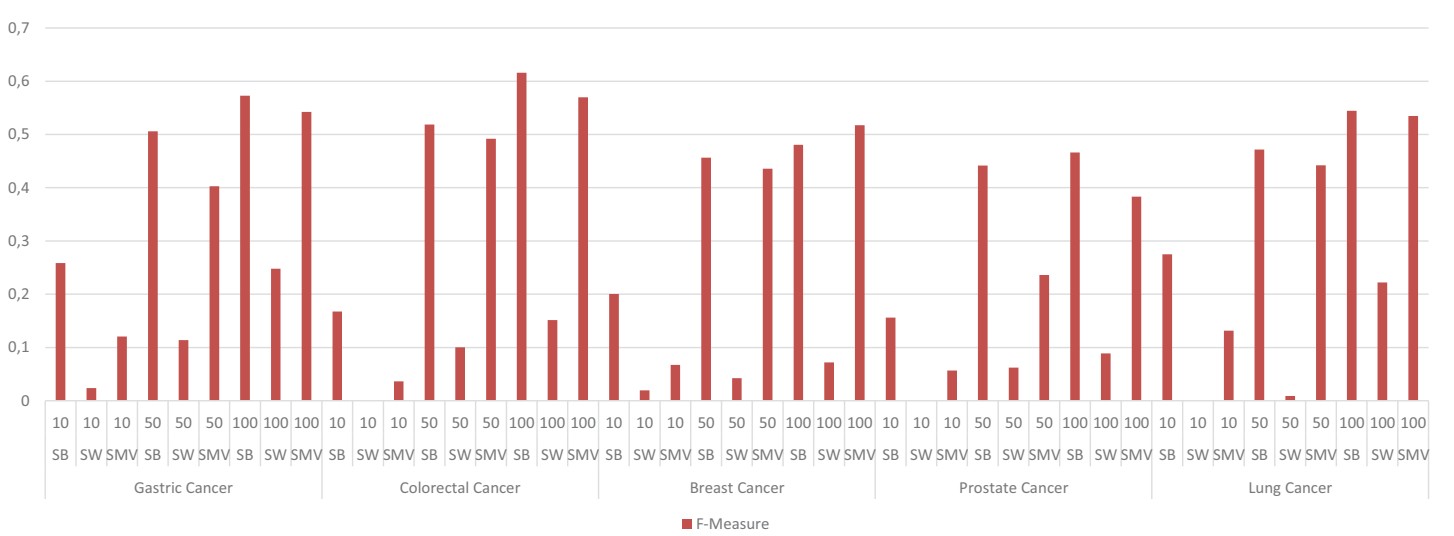

**Figure 8 F-measure values of single best (SB), single worst (SW) and SMV results by using GO-terms.**

measure values were retrieved by LSI and SMV based genes on all cancer types when the top ranked gene number was selected as 10. I also analyzed the effect of the ensemble approach by comparing the best and worst performance results of the LSI-based methods with the performance results of SMV by utilizing GO-terms in Fig. 8.

The F-measure values of SMV based gene lists varied between the highest and lowest F-measure values as expected. However, the highest results on breast cancer data was also achieved by SMV when the number of top ranked gene was set to 100. The performance results of SMV closely approached those of SB, particularly when the number of top-ranked genes was set to 100. These findings are important for emphasizing the efficiency of the ensemble approach. The increment in the selected genes in GSEA led to obtain more disease related GO-terms from the SMV based gene lists. The increment in

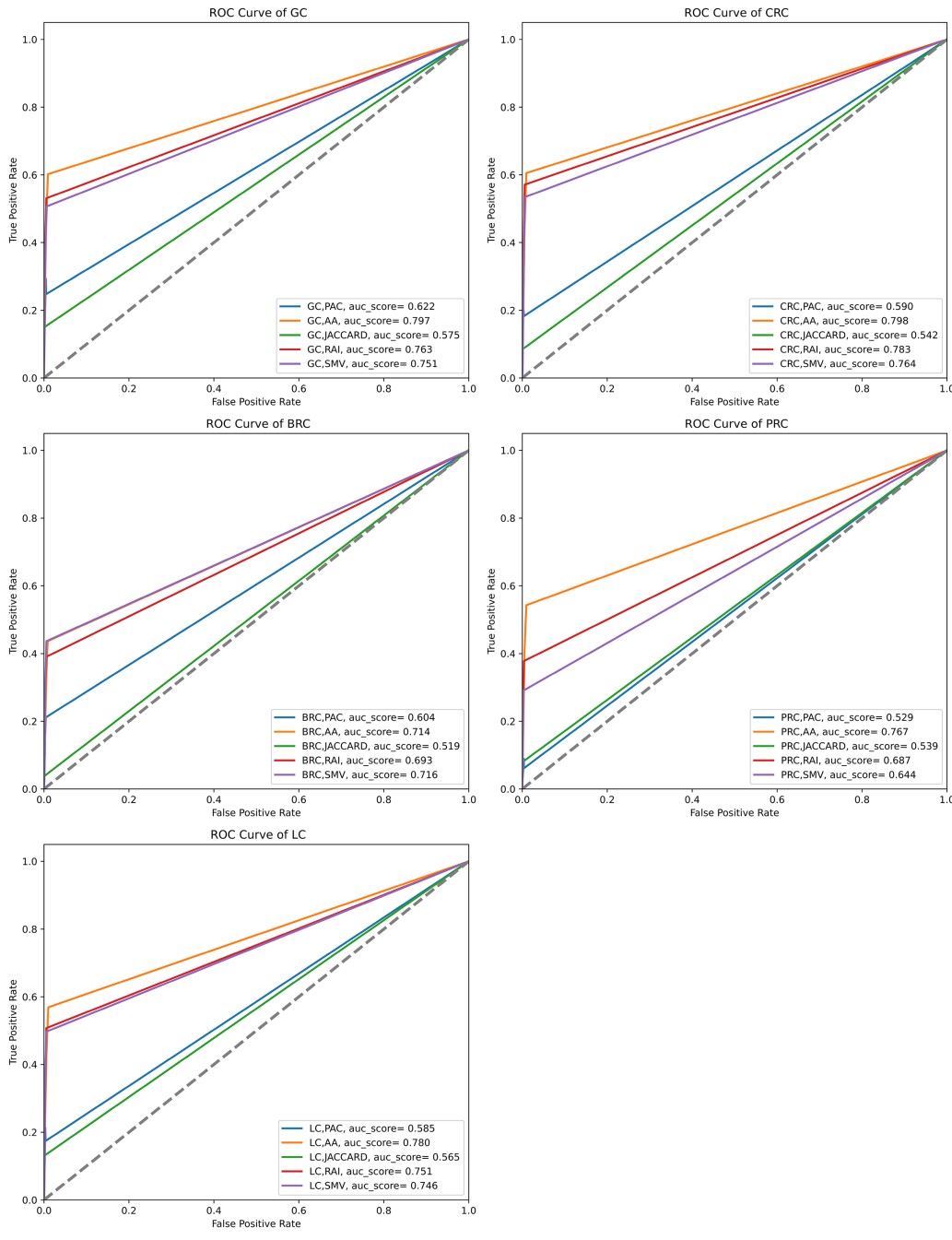

**Figure 9 ROC curve analysis of LSIs and SMV based inferred genes by GO-terms.**

the SMV based selected genes also increased gene diversity, which also enabled to retrieve the accurate and disease related GO-terms.

The performance of the overlap analysis between GO-terms of disease-related genes and GO-terms of LSIs and SMV based genes for each cancer type was also evaluated with ROC curves in Fig. 9. The top 10, 50, and 100 ranked genes were exploited to measure the performance of GO-terms of each inferred based gene list for five cancers. AA

outperformed on gastric, colorectal, prostate, and lung cancer data with 0.797, 0.798, 0.767, and 0.78 AUC scores. The performance of RAI on the same cancer types were also close to the performance of AA with 0.763, 0.783, 0.687, and 0.751. SMV performed the highest AUC score on breast cancer data. PAC and JC generally obtained significantly lower AUC scores than AA, JC, and SMV on all cancer types. When comparing the disease-related GO-terms obtained using the SMV-based gene lists with those derived from four LSI-based methods, it was observed that the SMV results closely approximated the highest AUC scores achieved by the LSI-based methods. The study concluded that SMV slightly enhanced the performance of inferred gene lists and it approximated the best individual LSI-based performance results. The AUC scores presented in Fig. 9 aligned closely with the F-measure values shown in Fig. 7, indicating consistent performance across these metrics.

## DISCUSSION

The identification of disease-related genes is a significant issue in bioinformatics. A single nucleotide mutation in genes may cause many diseases. However, some of the complex diseases such as cancers, diabetes mellitus can be related more than one specific gene. There can be various molecular relations, which disrupt the usual structure of biological functions and processes, can cause such complex diseases. With developments in sequencing technology, various biological datasets enable researchers to infer molecular relations to determine the disease related genes. However, GWAS are important for detecting disease related genes, they are time consuming and expensive.

The main objective of this study is to assess the effect of a simple ensemble approach, which combines the disease related gene predictions from various LSIs, on the accuracy of disease related gene prediction. The four local similarity indices were applied on HPRD to infer potential disease related genes of gastric, colorectal, breast, prostate and lung cancers. It also allowed to compare the performance of four LSIs based predictions on five different cancer types. The study intended to enhance performance of disease related gene prediction using ensemble method, SMV. The SMV algorithm aggregates all predictions of LSIs based predictions and if any predicted gene is inferred by at least half of four LSIs based predictions then it was determined as disease specific gene. The gene prediction for diseases were evaluated with two separate ways. First, the GeneCard dataset was exploited as validation dataset in the overlap analysis for evaluating the performance of LSI-based and SMV based gene lists. Second, disease related GO-terms were retrieved by using OMIM disease gene list in GSEA to measure how much GO-terms of LSI-based and SMV based gene lists were overlapped with disease related GO-terms.

LP studies in the literature intend to discover potential, missing and spurious links in complex networks. The study presented the performance results of LP prediction studies in gene prediction and complex networks in Table 2 to compare the performance results. The researchers of the study, prediction models, validation type for the performance evaluation, performance metrics, and the performance scores were presented respectively as columns in Table 2. The studies presented in Table 2 were selected based on the usage of similar similarity indices, network type, the usage of ensemble techniques for LP, or validation types to those employed in this study.

**Table 2 Performance comparison between literature and our study.**

| Researchers | Model | Network type | Validation type | Performance metrics | The highest performance score |
|---|---|---|---|---|---|
| *Ghiassian, Menche & Barabási (2015)* | Diffusion models | PPI network | OMIM & PheGenI (overlap analysis) | Recall, hit rate | 0.05–0.3 (recall) |
| *Liu et al. (2017)* | Local and global similarity indices | PPI network (Yeast, Human) and various complex networks | Overlap analysis (cross validation) | Precision | 0.062–0.853 (Yeast) 0.01–0.223 (Human) |
| *Martínez, Berzal & Cubero (2016)* | Local, global and quasi-local similarity | PPI network (Yeast) | Overlap analysis (cross validation) | Precision | 0–0.1857 |
| *Lee & Tukhvatov (2018)* | Local similarity | Social network | Overlap analysis | F-measure | 0.007658 (Adamic Adar) |
| *Zhang, Tong & Wu (2020)* | Ensemble local similarity indices | Yeast and five other complex network | Overlap analysis (cross validation) | Precision | 0.2586 (for Yeast) |
| *Chiu & Zhan (2018)* | LP with weak estimators | Three temporal networks | Classification | AUC score | 0.86–0.986 |
| *Wu et al. (2019)* | Ensemble local similarity indices | Eight real networks | Overlap analysis (cross validation) | Precision | 0.105–0.578 (varied from network to network) |
| Our study | Four LSI and integration of LSIs | PPI network (Human) | Overlap analysis | F-measure, AUC score | F-measure with GeneCard (GC, CRC, BC, PC, LC) [0.11, 0.054, 0.049, 0.02, 0.06] F-measure with GO-terms (GC, CRC, BC, PC, LC) [0.57, 0.61, 0.52, 0.47, 0.54] AUC scores with GeneCard (GC, CRC, BC, PC, LC) [0.54, 0.52, 0.514, 0.508, 0.524] AUC scores with GO-terms (GC, CRC, BC, PC, LC) [0.8, 0.8, 0.72, 0.77, 0.78] |

*Ghiassian, Menche & Barabási (2015)* applied diffusion models on PPI networks to predict genes associated to diseases; the recall performance of these genes was assessed by overlap analysis using the OMIM and PheGenI databases, and it was found to be between 0.05 and 0.3. *Liu et al. (2017)* and *Martínez, Berzal & Cubero (2016)* worked on PPI networks for LP *via* utilizing LSIs and global similarity indices. In *Liu et al. (2017)*, the precision results of the eight indices on Yeast PPI network varied from 0.062 to 0.853. The precision results of the eight indices on human PPI network, Figeys, were changed from 0.01 to 0.223. The global, local, and quasi-local indices' performance results on YST were found by *Martínez, Berzal & Cubero (2016)* to be extremely close following cross-validation, with precision values ranging from 0 to 0.1857. *Lee & Tukhvatov (2018)* found the highest F-measure value by employing Adamic-Adar indices on VKontakte which is equal to 0.0077. The proposed model in the study (*Zhang, Tong & Wu, 2020*) achieved 25.86% precision value; however, RAI performed 25.54% precision values, which was too close to the performance of the proposed model. *Chiu & Zhan (2018)* found AUC scores of

the proposed model on MathOverflow and Eu-core networks were close to the AUC scores of LSIs. However, the proposed model significantly increased the performance of individual LSIs on CollegeMsg, where AUC score of the proposed model is 0.855, however individual LSIs based AUC score was varied between 0.6 and 0.7. *Wu et al. (2019)* emphasized that ensemble approach slightly enhanced the precision scores, which varied between 0.105 and 0.578 on eight networks. However, the performance of individual LSIs and the ensemble approach were generally comparable across eight real world networks, the RA and CN outperformed on certain real networks among eight networks.

The results of the studies in the literatures presented that the performance of LSIs were close to results of more complex techniques in LP. RAI and AA obtained good results when comparing more complex algorithms in the literature and this study. PAC, AA, JC, and RAI were selected to prevent complexity problem for the LPs based integration. The selection of simple LSIs might decrease the diversity, however, the main target of the study, which was the investigation of the effect of a simple ensemble method for discovering disease related genes, was implemented easily.

The results of four local similarity metrics and their integration in this study were similar to the results from different studies. The performance results in the literature were also differentiated due to the evaluation process, where some of the studies exploited k-fold cross validation and others used the most related top n genes for performance assessment. GeneCard disease-gene relation dataset and disease related GO-terms were utilized as validation datasets in overlap analysis, thus cross validation wasn't needed in the study. In overlap analysis by using GeneCard, the highest F-measure values were obtained *via* the inferred gastric cancer genes by RAI as 0.11, the inferred colorectal cancer genes by JC as 0.054, the inferred breast cancer genes by SMV as 0.049, the inferred prostate cancer genes by PAC as 0.02 and the inferred lung cancer genes by PAC as 0.06. AA and RAI based generally outperformed on all five cancers and the lowest performance scores were retrieved by JC based similarity indices. However, the increment of the number of the top genes also increased the F-measure values, it reduced the precision values on some cancer types. The highest F-measure values were obtained when the top gene selection is higher, 100. The highest precision values were generally retrieved when the number of related genes is 10. The study also visualized the performance of the LSI-based and SMV based gene lists using ROC curves. The AUC scores of these ROC curves were similar to the F-measure values for both the LSI and SMV methods. However, the AUC scores for the LSI-based and SMV-based gene lists were close to 0.5, suggesting that the results were nearly equivalent to random predictions in the overlap analysis with GeneCard. SMV also could not significantly increase the disease related gene predictions based on AUC scores. However, the highest F-measure value was derived by SMV based gene list on breast cancer data.

The study also evaluated the biological performance of gene lists by deriving GO-terms with GSEA. It measured how well the GO-terms of the LSI-based and SMV based gene lists were overlapped to disease related GO-terms. The F-measures and AUC scores obtained from the biological evaluation using GO terms were significantly higher than those derived from the overlap analysis with the GeneCard dataset. RAI, AA, and SMV based gene lists

achieved more than 0.5 F-measure values in gastric, colorectal, and lung cancer. AUC scores of AA, RAI and SMV based gene lists were also higher than 0.75 for the same cancer types. The ensemble approach approximated the highest F-measure and AUC scores for all cancer types. SMV based gene list also outperformed on breast cancer data. The study can conclude that SMV enhanced the performance of inferred gene lists in the overlapping analysis with GO-terms. The results also emphasized that if the number of genes were increased, the performance of LSI-based and SMV based gene lists were also increased. When comparing the performance of the studies in the literature and the current study, the findings of the study clarified that the simple ensemble techniques have limited effects on the enhancement for the performance. The similar performance scores were obtained in this study and other studies in the literature.

The theorical contribution of the study investigated the ensemble learner affect in the link prediction. The integration of gene predictions of LSIs is not well studied area in link prediction. Therefore, the study intended to generate new insight to aggregate the gene predictions of four local similarity indices to infer disease related genes more accurately. For this purpose, a simple ensemble algorithm, SMV, was employed to retrieve diseases related genes. The performance of SMV was assessed against the four LSIs in the current study. To measure the effect of SMV based integration the overlap analysis with the GeneCard dataset, and also the biological assessment with GO-terms were utilized in the study. SMV based gene lists achieved good performance in the overlap analysis with GO-terms. The increment in the number of the top ranked genes also led to obtain better results using SMV methods on all cancer types according to both overlap and biological analyses. The diversity of the selected genes was also increased when the number of selected genes in SMV-based gene lists was high. The high diversity among inferred genes enabled to reveal more disease related GO-terms by SMV based gene lists.

There are also some limitations of the study. First, the four simple LSIs were applied to infer disease related gene lists. There are also outstanding link prediction algorithms, which may be global, quasi-local indices as similarity based indices, probabilistic and maximum likelihood based approaches (hierarchical random graph, local probabilistic model, *etc.*), dimensionality reduction approaches (embedding based, matrix factorization based, *etc.*) and other hybrid techniques. The usage of the four LSIs may lead to obtain similar gene predictions due to the lack of diversity. The selection of different approaches such as global, quasi-local similarity can increase the diversity in link prediction. The integration of different types of link prediction algorithms may increase the performance of the SMV due to the high diversity. For future work, different types of LP algorithms are employed to increase the diversity of the predictions. The more sophisticated ensemble decision approaches, such as weighted ensemble techniques or boosting-based algorithms, are also intended to be applied in future work. The usage of different PPI networks more than HPRD to apply LP algorithms may also increase the diversity in LP. The increment in the PPI network database selection may also enhance the performance of the disease related gene predictions in future work.

### Funding
The authors received no funding for this work.

### Competing Interests
The author declares that they have no competing interests.

### Author Contributions
- Mustafa Özgür Cingiz conceived and designed the experiments, performed the experiments, analyzed the data, prepared figures and/or tables, authored or reviewed drafts of the article, and approved the final draft.

### Data Availability
The code is available at GitHub and Zenodo:

- https://github.com/ozgurcingiz/LinkPredictiononBiologicalNetwork.

- ozgurcingiz. (2024). ozgurcingiz/LinkPredictiononBiologicalNetwork: LinkPredictiononBiologicalNetwork (version_1.0.0). Zenodo. https://doi.org/10.5281/zenodo.13323937.

- The Human Protein Reference Database (http://www.hprd.org) is a protein-protein interaction database for humans. This was active in the first half of 2024, though it was not accessible for a period of time in 2024. It can be accessed here if the original site is still unavailable: https://www.hsls.pitt.edu/obrc/index.php?page=URL1055173331.

The OMIM database from EnrichR is available at: https://maayanlab.cloud/Enrichr:OMIM_Disease.

### Supplemental Information
Supplemental information for this article can be found online at http://dx.doi.org/10.7717/peerj.17975#supplemental-information.

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
