# Peer review of "Ensemble decision of local similarity indices on the biological network for disease related gene prediction"

_PeerJ, doi:10.7717/peerj.17975_

## Round 0.1 · original submission · Major Revisions

The reviewers have raised several critical issues with your study, especially about the validity of your findings (validation of results is confusing and insufficient for comparative analysis), biases introduced by data and methodological choices. These issues suggest that the manuscript requires significant revisions in presentation, clarity of contributions, justification of methodological choices, and enhancement of validation techniques to ensure the robustness and credibility of its findings.

**Language Note:** The review process has identified that the English language must be improved. PeerJ can provide language editing services - please contact us at [email protected] for pricing (be sure to provide your manuscript number and title). Alternatively, you should make your own arrangements to improve the language quality and provide details in your response letter. – PeerJ Staff

Reviewer 1 ·

Basic reporting

The manuscript is well-written, but there are some formatting issues that need to be addressed.

1.Pay attention to capitalization, such as "Common Neighbors (CN), Jaccard Index (JC), Adamic-Adar (AA), Preferential Attachment (PAC), Resource Allocation Index (RAI), Sorensen Index, Salton Index, CAR-based indexes, Hub Promoted Index, Leicht-Holme-Newman Local Index are some commonly used local similarity indices" (Line 43-45).

2.Abbreviations that have already been defined do not need to be defined again, such as "Genome-wide association studies (GWAS)," please double-check (Line 67 and Line 102).

Experimental design

The experimental design is reasonable and exhibits a certain degree of novelty.

Validity of the findings

The experimental results provide sufficient data support and effectively explain the research objectives.

Reviewer 2 ·

Basic reporting

The overall organisation of the paper could be significantly improve. There are important references that are missing. For example in this paper, "Deep Learning for Link Prediction in Dynamic Networks Using Weak Estimators" the authors have used a similar approach for combining weak predictors to predict missing links. It is not clear how the proposed method compares to other state of the art.
What are the major contributions of this work? This should be clearly highlighted.

Experimental design

I am concerned about that novel contributions of this work. The authors have combined four popular similarity indices including PA, RA, JC, AA to predict missing links. What made them choose these indices out of others? There is no theoretical contribution of this work.

Validity of the findings

The validation results are not convincing. Table 2 is very confusing. It does not give a clear idea about how the proposed method compares with other methods.

Additional comments

Overall, the paper have limited contributions and the experimental evaluations are not sufficient.

Reviewer 3 ·

Basic reporting

No comment

Experimental design

Utilizing only four local similarity indices (LSIs) that are closely related, such as PAC, AA, JC, and RAI, may not be the best approach since four LSIs are based on similar principles of common neighbors and node degrees, potentially leading to redundant or correlated predictions. Local LSIs focus on immediate neighborhoods, failing to account for higher-order topological features and global network patterns that could be relevant for disease gene prediction.

Validity of the findings

Using the expanded OMIM database to validate the performance of the link prediction (LP) and simple majority voting (SMV) approach could potentially introduce some circularity or bias, given that the expanded OMIM database itself is generated by incorporating protein-protein interaction (PPI) networks (Enrichr : Avi Ma'ayan et al, 2013)
If the expanded OMIM database has been created by leveraging similar PPI network information or algorithms, then using it as a validation set for the LP/SMV approach may not provide an entirely independent and unbiased evaluation of the method's performance.
It would be more appropriate to use an independent validation set or gold standard dataset that has been curated and established through experimental evidence or expert curation, without relying on the same PPI network data or computational approaches used in the LP/SMV method.

Ghiassian et al paper mentioned in the reference also uses OMIM for validation but utilizes 70 different gene sets to validate the findings. To provide a more comprehensive and standardized evaluation of the link prediction and ensemble methods' performance, it is recommended to generate and analyze Receiver Operating Characteristic (ROC) curves. ROC curves can effectively visualize the trade-off between true positive rate and false positive rate across different decision thresholds, allowing for a more nuanced comparison of the methods' predictive capabilities

---

## Round 0.2 · Minor Revisions

• Some sections need editing for clarity and grammar. For example, "The one of the most important problem" should be corrected.
• The manuscript frequently uses "we" and "our" despite the author's note stating, "Mustafa Cingiz is the only author responsible for every part of the article." These should be replaced with "I" and "my" to reflect the single authorship. Otherwise, the author can write in a non-first-person language throughout.

Reviewer 2 ·

Basic reporting

The authors have addressed most of my concern in the revised manuscript.

Experimental design

The authors have addressed most of my concern in the revised manuscript.

Validity of the findings

The authors have addressed most of my concern in the revised manuscript.

Additional comments

The authors have addressed most of my concern in the revised manuscript.

---

## Round 0.3 · accepted · Accept

The author has adequately addressed all queries raised.